# Contrasting Effects of Oxytocin on MK801-Induced Social and Non-Social Behavior Impairment and Hyperactivity in a Genetic Rat Model of Schizophrenia-Linked Features

**DOI:** 10.3390/brainsci14090920

**Published:** 2024-09-13

**Authors:** Daniel Sampedro-Viana, Toni Cañete, Paula Ancil-Gascón, Sonia Cisci, Adolf Tobeña, Alberto Fernández-Teruel

**Affiliations:** 1Medical Psychology Unit, Department of Psychiatry & Forensic Medicine, Institute of Neurosciences, Autonomous University of Barcelona, 08193 Bellaterra, Barcelona, Spain; daniel.sampedro@uab.cat (D.S.-V.); antoni.canete@uab.cat (T.C.); paula.ancil3@gmail.com (P.A.-G.); adolf.tobena@uab.cat (A.T.); 2Department of Life and Environmental Sciences and Center of Excellence for Neurobiology of Dependence, University of Cagliari, 09042 Cagliari, Italy; sonia.cisci@yahoo.it

**Keywords:** RHA and RLA rats, schizophrenia, social interaction, hyperactivity, oxytocin, NMDA receptor antagonist, MK801

## Abstract

Social withdrawal in rodents is a measure of asociality, an important negative symptom of schizophrenia. The Roman high- (RHA) and low-avoidance (RLA) rat strains have been reported to exhibit differential profiles in schizophrenia-relevant behavioral phenotypes. This investigation was focused on the study of social and non-social behavior of these two rat strains following acute administration of dizocilpine (MK801, an NMDA receptor antagonist), a pharmacological model of schizophrenia-like features used to produce asociality and hyperactivity. Also, since oxytocin (OXT) has been proposed as a natural antipsychotic and a potential adjunctive therapy for social deficits in schizophrenia, we have evaluated the effects of OXT administration and its ability to reverse the MK801-impairing effects on social and non-social behavior and MK801-induced hyperactivity. MK801 administration produced hyperlocomotion and a decrease in social and non-social behavior in both rat strains, but these drug effects were clearly more marked in RHA rats. OXT (0.04 mg/kg and 0.2 mg/kg) attenuated MK801-induced hyperlocomotion in both rat strains, although this effect was more marked in RHA rats. The MK801-decreasing effect on exploration of the “social hole” was moderately but significantly attenuated only in RLA rats. This study is the first to demonstrate the differential effects of OXT on MK801-induced impairments in the two Roman rat strains, providing some support for the potential therapeutic effects of OXT against schizophrenia-like symptoms, including both a positive-like symptom (i.e., MK801-induced hyperlocomotion) and a negative-like symptom (i.e., MK801 decrease in social behavior), while highlighting the importance of the genetic background (i.e., the rat strain) in influencing the effects of both MK801 and oxytocin.

## 1. Introduction

Impairments of social behavior constitute a major disability factor in schizophrenia. Social withdrawal (i.e., asociality) is one of the most important negative symptoms of schizophrenia that has been linked to longer and more debilitating prodromal periods (e.g., [1,2]). Negative symptoms in general, and asociality in particular, are resistant to currently available pharmacological treatments; thus, they are the focus of intense preclinical research with rodent models in an effort to find new and effective therapeutic approaches (e.g., [3,4,5,6,7,8]).

In rodent models, administration of N-methyl-D-aspartic acid receptor (NMDAR) antagonists (such as MK801–dizocilpine-, or phencyclidine), produce schizophrenia-like positive, negative, and cognitive symptoms. This makes NMDAR antagonist administration a useful tool for research on glutamate-related neurobiological mechanisms of the disorder and for testing treatments [9,10,11,12,13,14]. The administration of NMDAR antagonists to induce social withdrawal (asociality, an example of negative-like symptoms) and hyperlocomotion (positive-like symptoms) in rodents is widely used and is currently considered a valid pharmacological model of schizophrenia-related symptoms [9,11,15,16,17]. Furthermore, it has been shown that the impairments induced by the administration of NMDAR antagonists can be reversed by the administration of atypical antipsychotics (e.g., [18,19]).

The cross-species-conserved, hypothalamic neuropeptide oxytocin (OXT) has been proposed as a potentially natural antipsychotic, according to studies with rodent models and humans [20,21,22,23,24,25,26]. Some studies with rodents have found that OXT reverses NMDAR antagonist-induced social deficits [15,19,24,27]. Consistently, there is some evidence that patients with schizophrenia present alterations of the oxytocinergic system. Accordingly, OXT might improve social cognition/behavior and/or increase motivation for social interaction in these patients [20,22,23,25,28,29,30,31,32].

The Roman high-avoidance (RHA) rat strain/line exhibits, relative to its Roman low-avoidance (RLA) counterparts, a wide range of schizophrenia-relevant phenotypes (see [33,34,35]). For example, RHA rats show enhanced novelty-induced locomotor activity [36], impaired latent inhibition [37,38] and prepulse inhibition of the startle response (PPI), deficient spatial working memory [39,40,41,42], and poorer maternal/nesting and social preference behavior ([43,44,45]; for review see [33,35]). Remarkably, RHA rats also exhibit enhanced locomotor and mesolimbic (dopaminergic) sensitization following chronic administration of psychostimulant drugs, and many other relevant behavioral, pharmacological, and neurobiological phenotypes related to schizophrenia [33,34,35]. In addition, neuroanatomical and molecular studies have revealed that the RHA strain displays brain anomalies (e.g., in prefrontal cortical and hippocampal function) that are reminiscent of schizophrenia [33]. Interestingly, in this regard, it has been reported that OXT administration attenuates the PPI deficits shown by RHA rats, which in turn present a lowered expression of the *CD38* gene (which regulates OXT secretion) in the prefrontal cortex (PFC) relative to their RLA counterparts [46]. In addition, we have found in our previous study that the social withdrawal and hyperactivity induced by administration of the NMDAR antagonist MK801 are markedly enhanced in RHA rats relative to RLAs, and both MK801 effects are selectively attenuated by atypical antipsychotics in the former strain [47].

Taking into account the above findings the present study was aimed at exploring the effects of MK801 alone or following pretreatment with OXT on social interaction and locomotor activity in RHA vs. RLA rats. It was expected that (1) MK801 would decrease social behavior more markedly in RHA rats than RLAs; (2) RHA rats would display more marked hyperactivity after MK801 administration than RLAs; and (3) the attenuation of MK801-impaired social behavior and hyperactivity by OXT would be more marked in RHA rats than in their RLA counterparts.

## 2. Materials and Methods

### 2.1. Subjects

A total of 154 experimentally naive male RHA and RLA rats were used. All the animals came from the permanent colonies maintained at the laboratory of the Medical Psychology Unit, Department of Psychiatry and Forensic Medicine (Autonomous University of Barcelona, Bellaterra, Spain). Animals were approximately 3 months old at the beginning of the experiment with an average weight of 317.5 ± 27.0 g for RLAs and 333.6 ± 22.0 g (mean ± SD) for RHAs. They were housed in same-sexed pairs in Makrolon cages (50 × 25 × 14 cm) and maintained under a 12:12 h light–dark cycle (lights on 08:00 a.m.), with controlled temperature (22 ± 2 °C) and humidity (50–70%). They had water and food available ad libitum. The rats in each of the 12 experimental groups (see below) came from at least 10 different litters.

All testing was carried out between 9:00 and 13:30 h. All procedures were in accordance with the ethical and legal requirements of Spanish legislation on the “Protection of Animals Used for Experimental and Other Scientific Purposes” and the European Communities Council Directives (5/1995/Government of Catalonia, 214/1997/Government of Catalonia, Royal Decree 53/2013, 86/609/EEC, 91/628/EEC, 92/65/EEC, and Directive 2010/63/EU). The project, with reference number 3875-CEEAH, was reviewed and approved by the OH-CEEA-UAB on 27th of July 2017.

### 2.2. Social Interaction

The social interaction (SI) set-up test was adapted and implemented based on the methodology described by Gururajan et al. [19] (see Figure 1). Two transparent acrylic boxes (65 × 23 × 20 cm) were positioned facing each other at a distance of 12 cm to prevent physical contact between the animals. Each box had two 3 cm diameter holes, one at each end wall, corresponding to the “social” (facing the other box) and “non-social” holes (the latter facing an empty space opposite to the other box) (see Figure 1). The experimental room was illuminated by a dim red light in the ceiling. A video camera was placed around 1.5 m above the testing SI boxes and testing sessions were all recorded (see Figure 1). Rats’ behaviors were scored by two well-trained observers (r = 0.95 between observers) from a TV screen placed just outside the experimental room. The observer was blind to the strain and treatment conditions of the animals (i.e., other experimenters picked up each rat from its home cage and did the drug treatments).

**Scheme 1 brainsci-14-00920-sch001:**
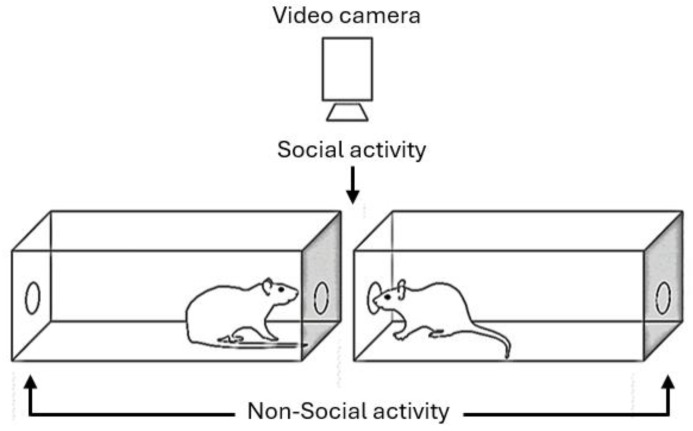
Schematic drawing of the experimental SI setup. The “social” holes are those that face the other box, and the “non-social” holes are the distal ones. This schematic was modified from the drawing in references [19,48].

One day before the behavioral SI test, a 30 min habituation session was carried out in which the two holes of each box were covered with tape, and a barrier was placed between the two boxes to prevent exploratory activity of the next box. A pair of weight-matched rats (maximum weight difference 25 g) were placed into the boxes (one rat in each box) for 30 min to familiarize themselves with the boxes and the experimental room. On the SI testing day, the barrier between the two boxes was removed, and different pairs of weight-matched unfamiliar animals were individually placed (one animal per SI box) in the set-up for 15 min with the holes uncovered, to allow exploration through them. The experimental room (in both the habituation and testing sessions) was dimly illuminated with a red light. After each habituation and SI testing session, each box was cleaned with 70% ethanol solution and dried with paper towel.

Variables measured in this experiment were: “Social time”, the time that a subject spent nose-poking at the social hole; “Non-social time”, the time spent nose-poking at the non-social hole; “Locomotor activity”, the number of crossings through the 3 sectors defined by lines painted with a marker on the floor of the testing boxes. Hole nose-poking, defined as a rat inserting its nose into a hole until the level of its eyes for at least 1 s, was measured using stopwatch.

### 2.3. Drug Treatment and Experimental Groups

RHA and RLA rats received a subcutaneous injection, 30 min before the start of the SI test, of sterile 0.9% saline vehicle, 0.04 mg/kg or 0.2 mg/kg of OXT (oxytocin 96%, J63421, Thermo Fisher Scientific, Waltham, MA, USA, dissolved in sterile 0.9% saline solution). Ten minutes after OXT (or vehicle) injection, i.e., 20 min before the SI task, rats received a second subcutaneous injection of MK801 (M107, Sigma-Aldrich; St. Louis, MO, USA) 0.15 mg/kg or sterile 0.9% saline vehicle.

Rats from each strain were randomly assigned to the 6 different experimental groups per strain (12 experimental groups in total): VEH-VEH, two vehicle injections; OXT0.04+VEH, oxytocin 0.04 mg/kg plus vehicle; OXT0.2+VEH, oxytocin 0.2 mg/kg plus vehicle; VEH+MK801, vehicle plus MK801 0.15 mg/kg; OXT0.04+MK801, oxytocin 0.04 mg/kg plus MK801 0.15 mg/kg; OXT0.2+MK801, oxytocin 0.2 mg/kg plus MK801 0.15 mg/kg.

All groups consisted of 12–14 rats, except RLA OXT0.2-MK group, in which n = 11 because 1 rat had to be excluded from analysis due to technical problems in the SI testing cage.

### 2.4. Statistical Analysis

The statistical analyses were carried out using the “Statistical Package for the Social Science” (SPSS, version 17).

To evaluate the effects of MK801 and OXT factors, and the interaction effects on the behavioral variables, factorial ANOVAs (2 “strain” × 2 “MK801” × 3 “OXT” levels) were performed. Post hoc Duncan’s tests for multiple comparisons were performed following ANOVAs’ significant interaction effects. For analyses, the “Social”, “Non-social” and “Total hole exploration” measures were Ln transformed to reduce variability and to improve equality of variances. The *p*-value threshold was set at *p* < 0.05.

## 3. Results

The results of the present study are presented in Figure 1a,b and Figure 2a,b.

Factorial ANOVA (2 “strains × 3 “OXT doses” × 2 “MK801 doses”) on “Social time” revealed “Strain” (F(1,141) = 8.99, *p* < 0.003), “MK801” (F(1,141) = 174.37; *p* < 0.001), “Strain × MK801” (F(1,141) = 24.37; *p* < 0.001) and “OXT × MK801” (F(2,141) = 6.05, *p* < 0.003) effects. These interaction effects reflect that MK801-induced impairment of social behavior is more marked in RHAs than in RLA rats (Figure 1a), whereas the MK801-reducing effect on social time is attenuated by OXT in RLA rats (Figure 1a; see post hoc Duncan’s tests).

Regarding “Non-social time”, the ANOVA revealed “MK801” (F(1,141) = 140.85, *p* < 0.001), “Strain × MK801” (F(1,141) = 22.69, *p* < 0.001) and “OXT × MK801” (F(2,141) = 6.39, *p* < 0.002) effects. The interactions indicate that MK801-induced decrease in “Non-social time” is more marked in RHAs than RLA rats (Figure 1b), whereas such an MK801 effect is attenuated by OXT in some cases, particularly with the OXT 0.04 mg/kg dose in both rat strains (Figure 1b; see post hoc Duncan’s tests).

Factorial ANOVA on “Total hole exploration” revealed “Strain” (F(1,141) = 8.34, *p* < 0.004), “OXT” (F(1,141) = 3.41, *p* < 0.04), “MK801” (F(1,141) = 172.59; *p* < 0.001), “Strain × MK801” (F(1,141) = 247.61; *p* < 0.001), and “OXT × MK801” (F(2,141) = 6.67, *p* < 0.002) effects. The interactions indicate that MK801 reduces “Total hole exploration” more markedly in RHA rats, while there is a trend for OXT to attenuate MK801 effects, although Duncan’s multiple range test does not detect differences between OXT-MK801 and VEH-MK groups in any strain (see post hoc Duncan’s test Figure 2b).

With regard to “Locomotor activity”, there were “Strain” (F(1,141) = 17.88, *p* < 0.001), “Oxytocin” (F(2,141) = 8.14, *p* < 0.001), and “MK801” (*F*(1,141) = 128.96, *p* < 0.001) effects, as well as a “Strain × MK801” (*F*(1,141) = 13.95; *p* < 0.001) interaction, indicating that MK801 induced more marked hyperactivity in the RHA rats compared with their RLA counterparts (Figure 2a). Also, an “OXT × MK801” (F(2,141) = 7.35; *p* < 0.001) interaction was observed, reflecting that OXT attenuates the effect of MK801 on locomotion. According to post hoc Duncan’s comparisons, such an attenuation of MK801-induced hyperlocomotion by OXT is more clearly observed in RHA rats (see post hoc Duncan’s test, Figure 2a).

## 4. Discussion

The present results show that MK801 decreases the time spent investigating both the social and non-social holes and total hole exploration, in RHA and RLA rats. Notably, these effects are more pronounced in the former strain. Such an enhanced MK801 effect in RHAs (relative to RLAs) is in agreement with our previous observations with a wide range of doses (0.05, 0.075, 0.1, 0.15, 0.2 mg/kg) of the drug [47]. Likewise, the results show a marked increase in locomotor activity in MK801-treated animals, which is also more pronounced in RHA than RLA rats, also in agreement with our previous studies [47,49]. The increased sensitivity of RHA rats to the effects of MK801 in the above behavioral measures gives support to the predictive and face validity of the RHA model as an analog of some schizophrenia-linked phenotypes.

It is worth mentioning that the behavior of the vehicle-treated RHA and RLA rats is in accordance with the study by Sampedro-Viana et al. [47], in the sense that there are no differences between both groups in social and non-social hole exploration, though there is a trend for higher exploration times in RHAs, particularly regarding non-social time. This leads to a slightly higher percentage of social hole preference (i.e., the ratio between social hole exploration and the sum of exploration time in both holes) in vehicle-treated RLAs relative to RHA rats (percentage mean ± SEM: RLA VEH-VEH, 55.2 ± 2.0; RHA VEH-VEH, 51.6 ± 0.6; Student’s t(26) = 1.73, *p* < 0.048, one-tailed). These results cohere with the difference in mean (±SEM) percentages of social hole preference observed between the VEH-VEH-treated RLA (61.1 ± 0.14) and RHA rats (54.7 ± 0.18; Student’s t(97) = 2.83, *p* < 0.01 two-tailed) in the study by Sampedro-Viana’s et al. [47]. As discussed in our previous reports, it is noteworthy that between completely naive (untreated, uninjected) RLA and RHA male rats, the former strain usually exhibits a 10–20% higher social preference ratio than the latter (i.e., the means of social preference usually ranging 60–70% in RLAs and 45–55% in RHAs) (see [44,45,47]). Such a difference falls to 4–8% in vehicle-injected RHA/RLA rats, which might be due to pre-test injection stress, as observed in the present study and already discussed by Sampedro-Viana et al. [47].

For the present analyses, however, we have not used the percentage of social hole preference, because, in the present study, the VEH-MK801-treated groups presented exploration times close to zero (somewhat lower than in the study by Sampedro-Viana et al. [47]), so that the denominator would be problematically low in these groups and thus very small variations of time spent in either hole would lead to disproportionate changes in the ratios (percentages) compared to the VEH-VEH and OXT-VEH groups. That is why we decided to analyze the time spent exploring the social and non-social holes rather than the ratios or percentages.

Some studies have suggested that OXT administration may improve social cognition/perception and some psychotic symptoms in patients with schizophrenia (e.g., [29,30,31] see Section 1). Along these lines, previous studies from our laboratory have shown that, at the doses used here, OXT was able to slightly but significantly improve PPI in RHAs (but not in RLAs) and in low-PPI-selected outbred HS rats, while inducing a parallel increase in the expression of the OXT receptor gene in the rats’ prefrontal cortex (PFC) [46]. Thus, it seemed reasonable to hypothesize that RHA rats would benefit more from OXT administration than RLAs in terms of attenuation of MK801-induced alterations in the SI test.

The present results show that oxytocin alone did not affect social behavior, total hole exploration, or locomotor activity in the SI test in RHA and RLA rats. This absence of effects has been replicated in our laboratory, with exactly the same results for both OXT doses. However, as said above, these two OXT doses modify PPI in RHAs and outbred rats and have central effects [46]. Previous studies in rodent models have shown pro-social effects of acutely administered (mainly intranasal) oxytocin (reviewed by [50]), although there have also been some reports of no effects of oxytocin alone on social interaction/preference in rodents and zebrafish (e.g., [26,51,52,53]). These contrasting findings might be attributed to differences in the doses, route of administration, species, and/or even the particular procedures used for testing social interaction.

Although oxytocin per se did not change social behavior nor locomotor activity, we report, for the first time in RHA and RLA rats, that OXT was able to significantly attenuate MK801 effects on activity and time spent investigating the social and non-social holes in the SI test. Thus, both OXT doses attenuated MK801-induced hyperactivity in RHA rats, whereas only the highest dose attenuated the hyperactivity of MK801-treated RLA rats. Conversely, the MK801-induced deficit of exploration of the social hole (social time) was significantly attenuated by both OXT doses only in RLA rats. In addition, the decreasing effect of MK801 on non-social hole exploration was only attenuated by the lowest OXT dose in both rat strains. It might seem that the reduction of MK801-induced hyperactivity by oxytocin plays an important role in the increase in exploration of the social and non-social holes. However, while this might be partly true, the present findings also suggest that the differential strain-related “OXT+MK801” effects on exploration of the social hole cannot be fully accounted for by the observed effects on locomotion, since “OXT + MK801” effects on locomotor activity are predominant in RHA rats but significant effects on “social time” are only observed in RLAs.

Previous reports have shown that OXT is able to attenuate or reverse the phencyclidine- and MK801-impairing effects on SI and PPI in rats and zebrafish (e.g., [26,27,54]), as well as PCP-induced hyperactivity in rats [24]. This suggests that the oxytocinergic system interacts with glutamatergic transmission to modulate some schizophrenia-linked phenotypes (e.g., [55,56]). The relevance and novelty of the present study lie in the following: (1) for the first time it extends the OXT-induced attenuation of the effects of NMDAR antagonism to the Roman rat strains, of which the RHAs model a range of schizophrenia-relevant features, including enhanced effects of MK801 (see Section 1, and review by [33]); (2) importantly, it shows that the extent to which OXT influences each specific MK801 effect (e.g., hyperactivity vs. decreased social investigation) depends on the rat strain (i.e., the genetic background); and (3) OXT effects on MK801-induced hyperactivity or decreased social behavior may not be parallel. That is to say, OXT may influence a particular effect of MK801 (e.g., hyperactivity in RHAs) but not the others (e.g., time in the social hole in RHAs), and the specific profile of OXT influence on MK801 effects may depend upon de genetic background, i.e., the specific rat strain.

Altogether, the present findings support a strain-dependent differential modulatory role of the oxytocinergic system after the blockade of NMDA receptors in RHA and RLA rats. This may be related to the reported alterations of central glutamatergic transmission and oxytocinergic function in RHAs compared with their RLA counterparts [46,57,58,59,60,61].

The present SI procedure has been behaviorally and psychopharmacologically validated using mainly males (e.g., [15,19,44,47]). Due to breeding and space limitations, we have not used female rats in the present study, although recent studies with naïve (untreated) Roman rats have shown some specific sex-related effects on social behavior [45]. Thus, it would be important to include females in future experiments with OXT in order to explore the generality of the present findings (and these from [47]) to both sexes. It could also be considered that the sample size might represent a limitation. However, it is worth noting that 12–14 animals per group, as used in the present study, is a larger “n” than the n ≤ 10/group that is commonly used in most pharmacological studies of this type with rodents. Therefore, it is reasonable to think that the present “n”/group, together with the complete factorial design combining strain and two “drug” factors (even with more than one dose in one case), gives the present study sufficient power.

The induction of social withdrawal by NMDAR antagonists (PCP, ketamine, MK801) is considered to be a valid model of social dysfunction observed in schizophrenia, whereas the hyperactivity induced by these NMDAR antagonists is considered to model some positive symptoms of the disorder [9,10,19,27,62]. Compared with RLA rats, the RHA model shows enhanced detrimental effects on social and non-social investigation after MK801 (0.15 mg/kg) administration, as well as much more pronounced MK801-induced hyperactivity (see also [47,49]). Previous studies have shown that these MK801-disrupting effects can be dose-dependently attenuated by various atypical antipsychotics predominantly in RHAs [47].

## 5. Conclusions

The present study shows that OXT reduces MK801-induced hyperactivity in RHAs (and to a lesser extent in RLAs), whereas OXT attenuates the MK801-induced deficit of social behavior in RLA rats. The disruption of non-social exploration by MK801 is, to a lesser extent, attenuated by OXT in both rat strains. Altogether, the present results give support to the notion that OXT may have moderate positive effects on some schizophrenia-relevant phenotypes in two rat strains that have not been tested before in the present procedure (i.e., OXT attenuation of MK801 disruption of social vs. non-social behavior and hyperactivity). Importantly, however, the present findings also indicate that the presence and/or magnitude of OXT effects on specific phenotypes may critically depend on the particular rat/rodent strain (i.e., the particular genetic background) being used.

## Data Availability

The data that support the findings of this study are available on request from the corresponding author. The data are not publicly available due to privacy restrictions.

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
