# Peer review of "Contrasting Effects of Oxytocin on MK801-Induced Social and Non-Social Behavior Impairment and Hyperactivity in a Genetic Rat Model of Schizophrenia-Linked Features"

_brainsci, 2024, doi:10.3390/brainsci14090920_

Round 1

Reviewer 1 Report

Comments and Suggestions for Authors

This study investigated the interaction between the effects of NMDA receptor antagonists and oxytocin in two genetic backgrounds. It is interesting, but there are a few points that need to be addressed.

Major points

(1)This study focuses on social behavior. In Fig 1, the trends of social time and non-social time in the RHA or RLA strains are approximately similar. Therefore, as the authors also describe, the effects of MK801 and oxytocin affect social as well as non-social behavior. If the paper is to be published with social behavior as the focus, including the title, the impact on social behavior should be shown in the ratio to non-social behavior or in the preference data.

Minor points

In Fig1 and 2

(1) The label on the figures should be the same as the label on the legends.

(2) It would be easier to read if the order of the explanatory notes in the graph were the same as the order in which the bars appear.

(3) What does Ln stand for? Do they need an abbreviation?

Reviewer 2 Report

Comments and Suggestions for Authors

1-)the expression of schizophrenic maybe rude. you may consider paraphrasing the word in the abstract( schizophrenia-like ).

2-)mention that in your previous study.

 In addition, we have found that the social withdrawal and hyperactivity induced by administration of the NMDAR antagonist MK801 are markedly enhanced in RHA rats relative to RLAs, and both MK801 effects are selectively attenuated by atypical antipsychotics in the former strain [47].

3-)you should write non-abbreviated version of h first.

All testing was carried out between 9:00 and 13:30h.

4-)you may avoid non-English sentences

Generalitat de Catalunya

5-)the sentence should consists of verb. you should clearly explain what is modified from.

 Modified from [19, 62].

6-)avoid non-abbreviated words.

Hole nose-poking, i.e., exploration of a hole, was measured by a stop-watch, and was defined as a rat intro- 139
ducing its nose in a hole until the level of its eyes for at least 1 s.

7-)you can mention novelty of study in the abstract briefly.

8-)you can further mention limitations of your study and suggest ways to improve it.

9-)you can mention it as one of the limitations of your study. they may make mistakes.

Rats’ behaviors were scored by two well-trained observers (r = 0.95 between 116 observers) from a TV screen placed just outside the experimental room.

10-)also you can mention sample size as one of the limitations of your study.

Round 2

Reviewer 1 Report

Comments and Suggestions for Authors

No specific comments.